# *Cinnamomum migao* H.W. Li Ethanol-Water Extract Suppresses IL-6 Production in Cardiac Fibroblasts: Mechanisms Elucidated via UPLC-Q-TOF-MS, Network Pharmacology, and Experimental Assays

**DOI:** 10.3390/cimb47100798

**Published:** 2025-09-26

**Authors:** Yuxin Lu, Yaofeng Li, Can Zhu, Mengyue Guo, Xia Liu, Xiangyun Chen

**Affiliations:** School of Basic Medicine, Guizhou University of Traditional Chinese Medicine, Guiyang 550025, China; 18685817028@163.com (Y.L.); lyfengcxy2010@163.com (Y.L.); zhucan056@gzy.edu.cn (C.Z.); 19912820157@163.com (M.G.); liuxia0851@126.com (X.L.)

**Keywords:** *Cinnamomum migao* H.W. Li, cardiac fibroblasts, IL-6, laurolitsine, hecogenin, ADRB2/JNK/c-Jun signaling pathway

## Abstract

This study aims to elucidate the active components and underlying molecular mechanisms by which the ethanol-water extract of *Cinnamomum migao* H.W. Li (MG-EWE) inhibits cardiac fibroblast (CF) transdifferentiation and IL-6 production, providing insights into its anti-myocardial fibrosis effects. The chemical composition of MG-EWE was characterized using UPLC-Q-TOF-MS. Network pharmacology analysis identified active constituents and their mechanisms in inhibiting IL-6 production in CFs. An isoproterenol (ISO)-induced rat CF model was established to evaluate the effects of MG-EWE and its main monomers, Laurolitsine and Hecogenin, on cell proliferation, migration, collagen metabolism, IL-6 production, and key proteins in the ADRB2/JNK signaling pathway. A total of 173 compounds were identified in MG-EWE, with 14 core constituents regulating IL-6 synthesis via 16 key targets, including ADRB2 and MAPK9. Gene Ontology enrichment indicated that MG-EWE affects phosphorylation and the JNK signaling cascade. Molecular docking showed strong binding affinities between Laurolitsine, Hecogenin, and their targets ADRB2 and JNK. Experimentally, MG-EWE, Laurolitsine, and Hecogenin significantly inhibited ISO-induced CF proliferation, migration, and hydroxyproline synthesis, as well as the expression of p-ADRB2, p-JNK, p-c-Jun, and IL-6. MG-EWE inhibits CF transdifferentiation and IL-6 production via the ADRB2/JNK/c-Jun signaling axis, mediated by its constituents Laurolitsine and Hecogenin, highlighting its potential for drug development targeting myocardial fibrosis.

## 1. Introduction

Myocardial fibrosis (MF) is a pivotal pathological process involved in a range of cardiac conditions, including heart failure, myocardial infarction, and arrhythmias [1]. Epidemiological data worldwide reveal that approximately 25% of patients with cardiovascular disease exhibit notable myocardial fibrosis lesions. The progressive worsening of these fibrotic lesions significantly increases the risk of adverse cardiovascular events and contributes to overall mortality [2]. The hallmark pathological features of MF are marked by the abnormal accumulation and spatial reorganization of extracellular matrix (ECM) components, especially collagen types I and III. This process results in increased myocardial stiffness, decreased ventricular compliance, and heightened electrical conduction heterogeneity, ultimately leading to irreversible cardiac dysfunction [3]. Central to this pathological cascade is the phenotypic transformation of cardiac fibroblasts (CFs) into myofibroblasts (MyoFbs), which underpins MF development [4]. Activated MyoFbs promote pathological ECM remodeling by expressing high levels of α-smooth muscle actin (α-SMA), developing extensive endoplasmic reticulum networks, and secreting excessive matrix proteins [5]. Importantly, MyoFbs not only contribute to excessive ECM deposition at sites of injury but also induce reactive fibrosis in remote myocardial areas through paracrine mechanisms, establishing a “fibrosis diffusion” effect [6]. Consequently, targeting the transdifferentiation of CFs into MyoFbs may serve as a strategic approach for early intervention, aiming to prevent or delay the onset and progression of myocardial fibrosis and subsequent cardiac dysfunction.

Interleukin-6 (IL-6) is a multifunctional cytokine critically involved in inflammation responses and fibrotic processes. It plays a significant role in the pathophysiology of various cardiovascular diseases [7]. Prior research indicates that CFs can promote their own transdifferentiation via IL-6 production and also exert paracrine effects on cardiomyocytes and other cardiac cell types, thereby exacerbating myocardial fibrosis [8]. IL-6 derived from CFs contributes to fibrosis through two primary mechanisms: first, it directly stimulates abnormal collagen synthesis; second, it induces CFs to transdifferentiate into MyoFbs. Moreover, transdifferentiated MyoFbs can feedback to amplify IL-6 secretion, establishing a self-perpetuating cycle of inflammation and fibrosis [9,10]. Clinically, elevated serum IL-6 levels in MF patients correlate positively with increased collagen volume fraction (CVF). Additionally, monoclonal antibody therapies targeting the IL-6 receptor have demonstrated efficacy in reducing myocardial fibrosis in animal models [11]. Therefore, inhibiting IL-6 production by CFs is a promising therapeutic strategy to prevent CF transdifferentiation and slow the progression of myocardial fibrosis.

Traditional Chinese medicine has demonstrated distinct advantages in the treatment of myocardial fibrosis, particularly due to its multi-target, multi-pathway holistic regulatory properties, which provide a novel perspective for reversing complex fibrotic networks [12]. *Cinnamomum migao* H.W. Li (MG) is the dried fruit of a plant within the Cinnamomum genus of the Lauraceae family. As an endemic medicinal plant of the karst regions in southwest China, it has traditionally been used to treat conditions such as coronary heart disease, angina pectoris, and asthma [13,14]. In Miao medicine, it is believed to have the functions of detoxification (referred to in Chinese as Jiedu), promoting blood circulation (referred to in Chinese as Huoxue), and facilitating the flow of Qi (referred to in Chinese as Tongqi) [15]. Modern chemical composition studies have shown that MG is abundant in a variety of bioactive compounds. These include vicinal diol sesquiterpenes and cinnamigones D-H [16,17], which exhibit neuroprotective effects, as well as miganoids A-J and 7(S)-(hydroxypropanyl)-3-methyl-2-(4-oxopentyl) cyclohex-2-en-1-one [18], which demonstrate anti-inflammatory activity. Modern pharmacological research indicates that MG can attenuate isoproterenol (ISO)-induced myocardial injury and alleviate symptoms of heart failure in rat models [15,19]. Furthermore, studies have identified novel sesquiterpenoid compounds isolated from this plant, which exhibit inhibitory effects on pro-inflammatory cytokines TNF-α, IL-1β, and IL-6. Notably, the expression of *IL-6* mRNA is significantly suppressed, suggesting its potential therapeutic role in modulating the inflammation-fibrosis axis [19]. However, it remains unclear whether MG and its active constituents can intervene in IL-6 production from CFs and inhibit CF transdifferentiation and what mechanisms are involved in this.

This study employed Ultra-performance liquid chromatography coupled with quadrupole time of flight mass spectrometry (UPLC-Q-TOF-MS/MS), network pharmacology, and cellular experiments to elucidate the molecular mechanisms by which MG inhibits CF transdifferentiation and IL-6 production. This study establishes a contemporary scientific foundation for understanding the anti-fibrotic effects of MG, while also identifying potential molecular candidates and providing theoretical groundwork for the development of new anti-fibrotic therapeutics that specifically target the IL-6 signaling network.

## 2. Materials and Methods

### 2.1. Reagents

MG was collected in Luodian County, Guizhou Province, China. The authenticity of the plant was verified by Professor Li Xuzhao from Guizhou University of Traditional Chinese Medicine. The collected sample was deposited at the Basic Medical Science Laboratory Center of Guizhou University of Traditional Chinese Medicine (catalog number: 202103MG). Rat Cardiac Fibroblast Cells (CP-R074) and Rat Cardiac Fibroblast Cell Complete Medium (CM-R074) were purchased from Wuhan Procell Life Science & Technology Co., Ltd. (Wuhan, Hubei, China). Isoprenaline hydrochloride (HY-B0468), Laurolitsine hydrochloride (HY-N2352A), Hecogenin (HY-N1422), and Propranolol hydrochloride (HY-B0573) were purchased from MedChemExpress Co., Ltd. (Monmouth Junction, NJ, USA). Crystal violet (71012314) was obtained from Sinopharm Chemical Reagent Co., Ltd. (Beijing, China). The cell Counting Kit (CCK-8 Kit) (HYCCK8) was purchased from HYCEZMBIO (Wuhan, Hubei, China). Rat Interleukin-6 (IL-6) ELISA Research Kit (MM-0190R1) was purchased from Jiangsu Meimian Co., Ltd. (Yancheng, Jiangsu, China). Hydroxyproline (Hyp) Assay Kit (A030-1-1) was purchased from Nanjing Jiancheng Bioengineering Institute (Nanjing, Jiangsu, China). HiScript^®^ II Q RT SuperMix for qPCR (+gDNA wiper) (R223-01) purchased from Vazyme Biotech Co., Ltd. (Nanjing, Jiangsu, China). The following primary antibodies were used in the study: Collagen I (139 kDa, ab260043) and Collagen III (139 kDa, ab184993) from Abcam Shanghai Trading Co., Ltd. (Shanghai, China); α-SMA (42 kDa, BF9212) from Affinity Biosciences (Changzhou, Jiangsu, China); ADRB2 (46 kDa, HY-P81085) from MedChemexpress Co., Ltd. (Monmouth Junction, NJ, USA); p-ADRB2 (46 kDa, AF3117) from Affinity Biosciences (Changzhou, Jiangsu, China); JNK (42, 50 kD, 66210-1-Ig) and p-JNK (42, 50 kDa, 80024-1-RR) from Proteintech Group, Inc. (Wuhan, Hubei, China); c-Jun (43 kDa, A11378) and p-c-Jun (43 kDa, 3270T) from ABclonal Biotechnology Co., Ltd. (Wuhan, Hubei, China) and Cell Signaling Technology, Inc. (Danvers, MA, USA), respectively; IL-6 (30 kDa, BA4339) from BOSTER Biological Technology Co., Ltd. (Wuhan, Hubei, China).

### 2.2. Preparation of Ethanol-Water Extract of MG

A total of 2 kg of MG was initially ground, sieved and accurately weighed. The powdered material was then immersed in 8 times its volume of 75% (*v*/*v*) ethanol and allowed to soak for 7 h. Following the soaking period, ultrasonic extraction was carried out for 80 min to enhance the release of active components. The resulting mixture was subsequently centrifuged at 13,000 rpm for 10 min to separate the liquid and solid phases. The supernatant was carefully collected, filtered to remove any remaining particulates, and set aside for further processing. The remaining residue was then subjected to two additional extraction cycles using 8 times its volume of deionized water, with each extraction lasting 2 h. After each water extraction, the mixture was filtered, and the filtrates were collected and set aside. The three extracted liquids—namely, the ethanol extract and two water extracts—were combined, and the combined solution was concentrated using a Rotary Evaporator (Shanghai Bilon Instrument Co., Ltd., Shanghai, China, BILON-T-300) under reduced pressure. The concentrated extract was then subjected to freeze-drying until a constant weight was achieved, ultimately yielding 145.2 g of MG ethanol-water extract (MG-EWE) in the form of a freeze-dried powder.

### 2.3. UPLC-Q-TOF-MS/MS Analysis

0.5 g MG-EWE lyophilized powder was dissolved in 10 mL of acetonitrile-water (containing 0.1% formic acid) solvent. A 5 μL aliquot of the resulting solution was injected for analysis. Chromatographic separation was carried out a Waters UPLC BEH C18 column (2.1 × 100 mm, 1.7 μm), with a flow rate of 0.4 mL/min and the column temperature maintained at 40 °C. The mobile phase was composed of 0.1% formic acid in water (solvent A) and acetonitrile containing 0.1% formic acid (solvent B), and gradient elution was employed. The gradient elution program was set as follows: 0–3 min, 5% to 15% B; 3–6 min, 15% to 30% B; 6–7 min, 30% B (isocratic); 7–12 min, 30% to 70% B; 12–15 min, 70% B (isocratic); 15–18 min, 70% to 100% B; 18–25 min, 100%B (isocratic). For mass spectrometry an AB 5600 Triple TOF (AB Sciex, Redwood City, CA, USA, Triple TOF^TM^ 5600+) mass spectrometer was used. The electrospray ionization (ESI) source parameters were configured as follows: nebulizer gas pressure (GS1): 55 psi, auxiliary gas pressure: 55 psi, curtain gas pressure: 35 psi, temperature: 550 °C, spray voltage: 5500 V (positive ion mode) or −4000 V (negative ion mode). Collision energy: 40 eV, collision energy difference: 20 V. The MS scan range is mass-to-charge ratio (*m*/*z*) 100–1200, and the MS/MS scan range is mass-to-charge ratio (*m*/*z*) 50–1200. The raw mass spectrometry data were imported into Waters Progenesis QI software 2.4v, and component identification of MG-EWE was performed by combining the data with the NIST and METLIN MS/MS databases using the Progenesis QI software.

### 2.4. Network Pharmacology

These compounds were entered into the PubChem database (https://pubchem.ncbi.nlm.nih.gov/ (accessed on 16 October 2024)) for retrieval and subsequently filtered based on Lipinski’s Rule of five as the screening criteria (MW ≤ 500; Hacc ≤ 10; Hdon ≤ 5; AlogP < 5; RBN ≤ 10) to identify candidate compounds. The target proteins associated with these candidate compounds were obtained from the PubChem database, the Traditional Chinese Medicine System Pharmacology Database and Analysis Platform (TCMSP, https://www.tcmsp-e.com/load_intro.php?id=43 (accessed on 19 October 2024)), and the SwissTargetPrediction website (http://www.swisstargetprediction.ch/ (accessed on 19 October 2024)). To explore the mechanisms involved in IL-6 production, relevant signaling pathways were identified through a search of the KEGG Pathway database. In parallel, the PubMed database (https://pubmed.ncbi.nlm.nih.gov/ (accessed on 20 October 2024)) was systematically searched using the keywords “Cardiac Fibroblast” and “IL-6”. Pathways that were either disease-specific or restricted to particular cell types were excluded from further analysis. Ultimately, the proteins directly associated with IL-6 production by CFs were selected as the key targets of interest.

The target proteins of the candidate compounds and the proteins involved in IL-6 production by CFs were uploaded to the Venny 2.1.0 online platform (https://bioinfogp.cnb.csic.es/tools/venny/index.html (accessed on 22 October 2024)) to determine the overlapping targets. The intersection targets were inferred to be the potential action targets through which the active components of MG-EWE may inhibit IL-6 production by cardiac fibroblasts.

These potential action targets were further analyzed by importing them into the STRING database (https://cn.string-db.org/ (accessed on 22 October 2024)) to conduct protein–protein interaction (PPI) analysis. Additionally, the DAVID database (https://davidbioinformatics.nih.gov/ (accessed on 22 October 2024)) was employed to perform Gene Ontology (GO) and Kyoto Encyclopedia of Genes and Genomes (KEGG) enrichment analyses on the potential action targets. the results of these enrichment analyses were visualized using the Microbiological Information Platform (https://www.bioinformatics.com.cn/ (accessed on 22 October 2024)). Finally, a network illustrating the relationships between the “MG-EWE active components and their potential action targets” was constructed using Cytoscape_v3.9.1 software.

### 2.5. Molecular Docking

The 2D structural information (in sdf format) of the compounds was retrieved from the PubChem database as small molecule ligands. The 3D structure (in PDB format) of the target proteins were obtained from the Protein Data Bank (PDB, https://www.rcsb.org/ (accessed on 28 October 2024)) as the protein receptors. The structures of small-molecule compounds were optimized using ChemDraw 22.0.0 64-bit software, while the protein structures were refined using PyMOL-3.1.1 software. The optimized structures of both the small molecule compounds and proteins were subsequently uploaded to the CB-Dock2 website (https://cadd.labshare.cn/cb-dock2/php/index.php (accessed on 31 October 2024)) for molecular docking analysis. Docking results with a vina score ≤ −8 kcal/mol were considered reliable and selected for further investigation. Finally, the molecular docking interactions were visualized and analyzed using PyMOL-3.1.1 software to interpret the binding modes and interactions between the ligands and receptors.

### 2.6. Cell Culture and Model Establishment

Rat CFs were employed for this study. The cell were cultured in Rat Cardiac Fibroblast Cell Complete Medium and maintained in a humidified incubator set at 37 °C with 5% CO_2_. When the cell monolayer reached approximately 80% confluence, passaging was performed to prevent overgrowth. To induce fibrosis, the cells were treated with ISO at a final concentration of 10 μmol/L added directly to the culture medium [20]. Following treatment, the cells were incubated at 37 °C with 5% CO_2_ for an additional 48 h, during which the cells developed a fibrotic phenotype, thus establishing the in vitro cardiac fibroblast fibrosis model.

### 2.7. Drug Concentration Screening

#### 2.7.1. CCK8 Assay

First, to assess the in vitro cytotoxicity of MG-EWE and its active constituents (Laurolitsine and Hecogenin), rat cardiac fibroblasts (CFs) were plated in a 96-well plate at a density of 5 × 10^3^ cells per well. Experimental groups were established, including a normal control group (Control) and treatment groups exposed to varying concentrations of Laurolitsine (1, 30, 60, 120, 240 μM), Hecogenin (1, 20, 40, 80, 160 μM), and MG-EWE (1, 2, 3, 4, 5 mg/mL). Following a 48 h continuous culture period, 10 μL of CCK8 reagent was added to each well, and the plates were incubated at 37 °C for an additional 4 h. Cell viability and drug-induced cytotoxicity were evaluated by measuring the absorbance at 450 nm using a Microplate Reader (Thermo Fisher Scientific, Waltham, MA, USA, Multiskan FC).

Second, to investigate the impact of these compounds on cell proliferation under ISO-induced conditions, CFs were assigned to three groups: a normal control group (Control), an ISO-induced model group (ISO Model), and drug intervention groups treated with different concentrations of Laurolitsine (0.25, 0.5, 0.75 and 1 μM), Hecogenin (0.25, 0.5, 0.75 and 1 μM), or MG-EWE (0.25, 0.5, 0.75 and 1 mg/mL). In the drug intervention groups, cells were co-treated with 10 μmol/L ISO. All groups were incubated for 48 h. After the incubation period, 10 μL of CCK8 solution was added to each well, followed by 4 h incubation at 37 °C, and the absorbance at 450 nm was measured using a microplate reader to determine cell proliferation.

#### 2.7.2. ELISA Analysis

The culture supernatants from each group of cells were collected, and 100 μL aliquots were used for analysis. The levels of IL-6 secreted by the cells were quantitatively measured in strict accordance with the manufacturer’s instructions provided in the IL-6 ELISA kit.

### 2.8. Experimental Grouping and Intervention

The cell experiments were divided into six groups: the normal control group (Control), the ISO-induced model group (ISO Model), the Laurolitsine combined with ISO group (Laurolitsine + ISO), the Hecogenin combined with ISO group (Hecogenin + ISO), the MG-EWE combined with ISO group (MG-EWE + ISO), and the Propranolol combined with ISO group (Propranolol + ISO). Laurolitsine, Hecogenin and MG-EWE were administered at their respective optimal concentrations, which had been previously determined in Section 2.7. With the exception of the normal control group, all other experimental groups were switched to culture medium that contained either ISO alone or ISO in combination with the respective test drugs. This drug-containing medium was maintained for a continuous intervention period of 48 h.

### 2.9. Detection of Hydroxyproline (Hyp) Content

The cell culture supernatant from each experimental group was carefully collected. The content of hydroxyproline (Hyp), a well-established product of collagen metabolism, was then quantitatively measured in strict accordance with the manufacturer’s instructions provided in the hydroxyproline (Hyp) detection kit.

### 2.10. Transwell Cell Migration Assay

Cells in the logarithmic growth phase were harvested and resuspended in serum-free DMEM medium. The cell suspension was adjusted to a final density of 2.5 × 10^5^ cells/mL. A Transwell chamber was placed into each well of a 24-well plate. The lower chamber of the Transwell system was filled with 600 μL of medium containing 10% FBS, while the upper chamber received 100 μL of the prepared cell suspension from each experimental group. For these groups undergoing drug intervention, the corresponding concentration of drug was added at the time of cell inoculation. After 24 h incubation period, the Transwell chambers were carefully removed. The cells were fixed using a 70% pre-cooled ethanol solution for 1 h, followed by staining with 0.1% crystal violet for 20 min. The cells were then washed with phosphate-buffered saline (PBS), and the number of migrated cells was visualized under a microscope, photographed, and subsequently counted.

### 2.11. Real-Time Fluorescence Quantitative PCR

Total RNA was extracted from cells in each group using 1 mL of Trizol reagent. The concentration and purity of the extracted RNA were assessed using a Spectrophotometer (Hangzhou Miu Instruments Co., Ltd., Hangzhou, China, ND-100). For reverse transcription, 3 μg of RNA was used under the following reaction conditions: 50 °C for 15 min, 85 °C for 5 s, and 4 °C for 10 min. The resulting complementary DNA (cDNA) was diluted 20-fold prior to analysis. The expression level of *IL-6* mRNA was quantified using Real-Time Fluorescence Quantitative PCR (qPCR) (Applied Biosystems Inc., Waltham, MA, USA, Quant Studio 6). The housekeeping gene β-actin was used as an internal reference. The relative expression level of *IL-6* mRNA was calculated using the 2^−ΔΔCt^ method. The Primers in this assay were designed with Primer Premier 6.0 software, and their sequences are presented in Table 1.

### 2.12. Western Blot Analysis

Total proteins were extracted from the cells following the protocol provided in the total protein extraction kit, and the protein concentration was determined using the bicinchoninic acid BCA method. For each sample, 40 μg of protein was loaded onto a sodium dodecyl sulfate-polyacrylamide gel (SDS-PAGE) for electrophoresis. The resolved proteins were then transferred onto a polyvinylidene difluoride (PVDF) membrane under a constant current of 300 mA. The membrane was blocked with 5% skim milk at room temperature to prevent nonspecific binding. Primary antibodies were incubated with the membrane at 4 °C for 12 h. The primary antibodies include: α-SMA (1:1000), Collagen I (1:1000), Collagen III (1:1000), ADRB2 (1:1000), p-ADRB2 (1:1000), JNK (1:10,000), p-JNK (1:2000), c-Jun (1:1000), p-c-Jun (1:1000), IL-6 (1:1000), and β-actin (1:10,000). After incubation with the primary antibodies, the membrane was washed three times with Tris-buffered saline containing Tween 20 (TBST) for 10 min each. Subsequently, the membrane was incubated with a horseradish peroxidase (HRP)-labeled secondary antibody (1:10,000) at room temperature for 2 h. Protein bands were visualized using enhanced chemiluminescence (ECL) and the resulting gray values were quantitatively analyzed using Image-Pro Plus 6.0 software.

### 2.13. Statistical Analysis

All experimental data are presented as the mean ± standard deviation (mean ± SD). Statistical analysis was conducted using GraphPad Prism 10.1.2. Prior to inter-group comparisons, the data were tested for normality using the Shapiro–Wilk method and for homogeneity of variance using the Bartlett method. Intergroup differences were assessed using one-way analysis of variance (ANOVA). A *p*-value of less than 0.05 was considered to indicate statistical significance.

## 3. Results

### 3.1. Identification of Chemical Components in MG-EWE

UPLC-Q/TOF-MS analysis was employed to characterize the chemical constituents present in MG-EWE (Figure 1). A comprehensive total of 173 compounds were successfully identified, comprising 87 compounds detected in positive ion mode and 86 compounds in negative ion mode (Appendix A). The identified compounds spanned multiple major categories, including alkaloids, phenolic compounds, carboxylic acids and their derivatives, cinnamic acids and their derivatives, coumarins and their derivatives, fatty acyls, flavonoids, as well as various other organic compounds. This diverse range of chemical classes underscores the complex and multifaceted chemical composition inherent to the MG-EWE extract.

### 3.2. Network Pharmacology Reveals That MG-EWE Inhibits IL-6 Production in Cardiac Fibroblasts via the ADRB2/JNK Pathway

A comprehensive screening process identified a total of 132 active components from MG-EWE using Lipinski’s Rule of Five as a filter. The potential biological targets of these components were predicted by integrating data from the PubChem, TCMSP, and SwissTargetPrediction databases, resulting in a pool of 888 potential target proteins. From the KEGG Pathway and PubMed databases, 74 proteins specifically associated with IL-6 production in cardiac fibroblasts were selected. The intersection of these two datasets yielded 28 overlapping proteins (Figure 2), which corresponded to 43 active compounds within MG-EWE. Molecular docking studies were then performed to evaluate the binding affinity between these compounds and their respective targets. The results demonstrated that 14 compounds exhibited significant binding affinity (with vina scores ≤ −8 kcal/mol) to 16 target proteins (Table 2). Based on these interactions, an MG-EWE active component–target network was constructed (Figure 3A). These findings collectively suggest that MG-EWE may exert its inhibitory effect on IL-6 production in cardiac fibroblasts by modulating 16 key target proteins through the action of 14 specific active components.

Further analysis using a protein–protein interaction (PPI) network (Figure 3B) revealed that the 16 identified target proteins are closely interconnected, indicating potential collaborative roles in regulating IL-6 production. Gene Ontology (GO) enrichment analysis (Figure 3C) highlighted that the biological processes (BPs) most significantly enriched among these targets included protein phosphorylation and JNK cascade reactions. Regarding molecular functions (MFs), the targets were primarily associated with ATP binding and MAP kinase activity. In terms of cellular components (CCs), the proteins were notably enriched in key intracellular regions such as the cytoplasm and nucleus. These functional enrichments closely aligned with the high-scoring key targets identified in the molecular docking analysis, namely ADRB2 and members of the JNK family (MAPK8/9/10). KEGG pathway enrichment analysis (Figure 3D) further identified the top two enriched signaling pathways as the Toll-like receptor (TLR) pathway and the mitogen-activated protein kinase (MAPK) signaling pathway. Key proteins involved in these pathways included MAP2K1, MAP2K2, CHUK, IRAK4, MAPK14, TNF, MAPK10, and MAPK11. Among them, the JNK family (MAPK8/9/10), as a central node within the MAPK signaling cascade, plays a pivotal role in phosphorylating and activating the transcription factor c-Jun, which in turn regulates the expression of IL-6. Molecular docking analyses provided additional evidence that Laurolitsine exhibited stable binding with ADRB2, while Hecogenin displayed high affinity for the JNK family proteins (MAPK8/9/10) (Figure 4, Table 2). Based on the aforementioned findings, MG-EWE is likely to exert its effects by targeting ADRB2 and JNK family proteins through its core bioactive constituents, Laurolitsine and Hecogenin. This targeted interaction may subsequently modulate the phosphorylation of c-Jun, a critical component of the AP-1 transcription factor complex, ultimately leading to the suppression of IL-6 synthesis in cardiac fibroblasts.

### 3.3. Screening of Optimal Intervention Concentration for MG-EWE and Its Active Components

To assess the cellular safety of MG-EWE, Laurolitsine, and Hecogenin, this study employed the CCK8 assay to evaluate the proliferation viability of cardiac fibroblasts (CFs) in each experimental group of rats. The experimental data demonstrated that when the concentration of Laurolitsine was 1 μM, the concentration of Hecogenin was 1 μM, and the concentration of MG-EWE was 1 mg/mL, there were no statistically significant differences in CFs cell viability compared to the Control group (Figure 5A–C), with cell survival rates in all groups exceeding 85%. Notably, as the concentrations of these compounds were further increased, the cell survival rate exhibited a significant dose-dependent decline, indicating that higher concentrations may induce cytotoxic effects. Based on this safety evaluation, the research team re-designed the concentration gradient scheme to establish a foundation for selecting the optimal working concentrations for subsequent experimental procedures.

In the ISO-induced transdifferentiation model of rat CFs, following intervention with varying concentration gradients of MG-EWE, Laurolitsine, and Hecogenin, the CCK8 assay results indicated that compared to the Control group, ISO stimulation significantly enhanced the abnormal proliferative activity of CFs. In contrast to the ISO model group, the treatment groups receiving 0.75 μM and 1 μM Laurolitsine, 1 μM Hecogenin, and 0.5 mg/mL, 0.75 mg/mL, and 1 mg/mL MG-EWE all demonstrated significant inhibitory effects on CFs proliferation (Figure 5D).

To assess the therapeutic efficacy of the compounds, this study utilized an enzyme-linked immunosorbent assay (ELISA) to quantitatively measure the levels of IL-6 in the cell culture supernatant of each group. The experimental results showed that compared to the Control group, the secretion of IL-6 in the cell culture medium of the ISO model group was significantly elevated. In comparison to the ISO model group, all drug intervention groups exhibited varying degrees of reduction in IL-6 content, and this inhibitory effect displayed a concentration-dependent trend. Notably, the most pronounced reduction in IL-6 content was observed in the treatment groups receiving 1 μM Laurolitsine, 1 μM Hecogenin, and 1 mg/mL MG-EWE (Figure 5E). Based on the aforementioned results, the following concentrations—Laurolitsine (1 μM), Hecogenin (1 μM), and MG-EWE (1 mg/mL)—which exhibited no significant cytotoxicity and optimal biological effects, were ultimately selected as the working concentrations for subsequent experimental investigations.

### 3.4. MG-EWE, Laurolitsine, and Hecogenin Reduce the Ability of ISO-Induced CFs to Secrete Hyp

Compared with the control group, the Hyp content in the cell culture supernatant of the ISO-induced model group was markedly elevated, demonstrating that ISO effectively promoted the transdifferentiation and activation of cardiac fibroblasts into myofibroblasts. In comparison to the ISO model group, the Hyp levels in the supernatant of the Hecogenin, Laurolitsine, MG-EWE, and propranolol treatment groups were significantly reduced (Figure 6). These findings suggest that MG-EWE, Laurolitsine, and Hecogenin can diminish the secretion of Hyp by rat CFs under ISO stimulation and inhibit their transdifferentiation into myofibroblasts, thereby potentially mitigating fibrotic processes.

### 3.5. Inhibitory Effects of MG-EWE, Laurolitsine, and Hecogenin on ISO-Induced Migration of Rat CFs

To assess the anti-fibrotic properties of MG-EWE, Laurolitsine, and Hecogenin, this study employed a Transwell migration assay to evaluate the migratory capacity of rat CFs in each group. As shown in Figure 7, compared with the Control group, the ISO model group exhibited a marked increase in the number of migrating cells observed within the microscope field. In contrast, treatment with Hecogenin, Laurolitsine, MG-EWE, and the positive control propranolol all resulted in a significant reduction in the number of migrated cells when compared to the ISO model group. These findings demonstrate that MG-EWE and its bioactive constituents, Hecogenin and Laurolitsine, can effectively suppress ISO-induced migration of CFs, thereby indicating their potential therapeutic role in attenuating cardiac fibrosis.

### 3.6. MG-EWE, Laurolitsine, and Hecogenin Reduce the Expression of Collagen I, Collagen III, and α-SMA Proteins in ISO-Induced Rat CFs

To explore the effects of MG-EW, Laurolitsine, and Hecogenin on the activation of CFs and the progression of myocardial fibrosis, this study employed Western blot analysis to assess the protein expression levels of Collagen I, Collagen III, and α-SMA. The experimental findings demonstrated that, in comparison with the Control group, the ISO Model group exhibited a significant upregulation in the expression of Collagen I, Collagen III, and α-SMA. In contrast, treatment with Hecogenin, Laurolitsine, MG-EWE, and Propranolol resulted in a marked downregulation of these proteins when compared to the ISO Model group (Figure 8). These findings suggest that MG-EWE, Laurolitsine, and Hecogenin effectively suppress the expression of Collagen I, Collagen III, and α-SMA, thereby inhibiting transdifferentiation of CFs into MyoFbs and attenuating the excessive deposition of ECM.

### 3.7. MG-EWE, Laurolitsine, and Hecogenin Reduce the Expression Level of IL-6 mRNA in ISO-Induced Rat CFs

To investigate whether MG-EWE and its active components Laurolitsine and Hecogenin can suppress *IL-6* expression in ISO-induced rat CFs, this study employed qRT-PCR to measure *IL-6* mRNA expression levels across different experimental groups. The results demonstrated that, compared to the Control group, the expression level of *IL-6* mRNA in the CFs of the ISO Model group was significantly increased. In contrast, treatment with Hecogenin, Laurolitsine, MG-EWE, and the positive control drug propranolol all led to a marked reduction in *IL-6* mRNA levels when compared to the ISO-induced group (Figure 9). These research results confirm that MG-EWE and its active components Laurolitsine and Hecogenin effectively inhibit the transcription of the *IL-6* gene in ISO-induced rat CFs.

### 3.8. MG-EWE, Laurolitsine, and Hecogenin Significantly Attenuate the Phosphorylation of ADRB2, JNK and c-Jun Proteins, as Well as the Expression of IL-6 Protein in ISO-Induced Rat CFs

The results of Western Blot analysis are presented in Figure 8A. Compared with the Control group, the ISO-induced group exhibited markedly elevated levels of phosphorylated ADRB2 (p-ADRB2), phosphorylated JNK (p-JNK), phosphorylated c-Jun (p-c-Jun), as well as the ratio of p-ADRB2 to total ADRB2, p-JNK to total JNK, p-c-Jun to total c-Jun, and the expression level of IL-6 protein in CFs. In contrast, treatment with Hecogenin, Laurolitsine, MG-EWE, or the positive control propranolol significantly reduced the levels of p-ADRB2/ADRB2, p-JNK/JNK, p-c-Jun/c-Jun, and IL-6 protein when compared to the ISO Model group (Figure 10A–D). These findings suggest that MG-EWE and its active components can suppress ISO-induced IL-6 production in CFs, potentially through modulation of the ADRB2/JNK/c-Jun signaling pathway.

## 4. Discussion

Myocardial fibrosis is predominantly driven by the activation of CFs, a pathological process characterized by the excessive and aberrant deposition of ECM proteins within the perivascular regions and myocardial interstitium. This leads to significant impairment of both cardiac structure and function [21]. The activation of CFs represents a central event in the progression of myocardial fibrosis, typically triggered by various stimuli such as inflammation and mechanical stress subsequent to cardiac injury [22]. Given the critical role of CFs activation in the pathogenesis of myocardial fibrosis, the present study was designed to investigate the molecular mechanisms through which MG-EWE and its bioactive constituents influence CFs transdifferentiation and IL-6 production.

In this investigation, UPLC-Q-TOF-MS technology was employed to conduct a qualitative analysis of the chemical constituents of MG-EWE, resulting in the identification of a total of 173 distinct chemical components. Subsequent network pharmacology and molecular docking analyses revealed that 14 potential active components within MG-EWE are capable of inhibiting IL-6 production by CFs, targeting a total of 16 potential protein targets. The constructed PPI network illustrated complex interconnections among these targets, indicating their likely involvement in an intricate and interconnected protein signaling network. Further enrichment analyses using GO functional annotation and KEGG pathway analysis demonstrated that the active components of MG-EWE predominantly modulate biological processes such as phosphorylation events and the JNK cascade, with a particular focus on the MAPK signaling pathway. Among the identified targets, ADRB2 and JNK exhibited favorable binding affinities with their corresponding active components, Laurolitsine and Hecogenin, respectively. These findings suggest that Laurolitsine and Hecogenin may serve as the principal bioactive constituents of MG-EWE responsible for its anti-myocardial fibrosis effects.

Previous research has established that Laurolitsine is a member of the aporphine alkaloid class and exhibits a broad spectrum of pharmacological properties, including anti-diabetic, anti-hyperlipidemic, anti-malarial, and anti-tumor activities. Its mechanisms of action are diverse, encompassing the activation of the AMPK signaling pathway, regulation of mitochondrial function, and modulation of the gut microbiota [23,24]. Similarly, Hecogenin is a steroidal sapogenin endowed with multiple biological effects, such as anti-inflammatory, analgesic, anti-tumor, antibacterial, antioxidant, anti-ulcer, and neuroprotective properties. The multi-target and multi-mechanism characteristics of Hecogenin render it a promising candidate in drug discovery [25]. In an ISO-induced myocardial infarction model, administration of Hecogenin was shown to significantly reduce serum levels of cardiac injury biomarkers, including creatine kinase-MB (CK-MB), cardiac troponin T (cTnT), and cardiac troponin I (cTnI). It also improved cardiac weight parameters and morphological indices, such as the heart weight/body weight ratio (HW/BW). The underlying mechanisms may involve the inhibition of the NF-κB signaling pathway, leading to reduced expression of pro-inflammatory cytokines (TNF-α, IL-1β, IL-6) and thereby attenuating myocardial inflammation. Furthermore, Hecogenin enhances the activity of endogenous antioxidant enzymes (SOD, CAT, GPx) and diminishes lipid peroxidation product levels, thereby mitigating oxidative stress-induced damage. Additionally, Hecogenin downregulates the expression of the pro-apoptotic protein p53, thereby reducing cardiomyocyte apoptosis and synergistically protecting cardiac structural integrity alongside the NF-κB pathway [26]. The findings of the current study demonstrate that MG-EWE, along with its active monomers Laurolitsine and Hecogenin, can markedly inhibit ISO-induced CFs proliferation, migration, and Hyp secretion. These effects suggest a potent suppression of the phenotypic transformation of rat CFs into MyoFbs, representing a significant novel finding that underscores the potential therapeutic value of MG-EWE and its active components in the context of anti-myocardial fibrosis.

During CFs transdifferentiation, factors such as Collagen I, Collagen III, and α-SMA play key roles. Alteration in the mechanical microenvironment serve as critical for the transformation of CFs into MyoFbs. Cardiac injury compromises tissue structural integrity, facilitating the transmission of mechanical stress to CFs. In conjunction with multiple cytokines, this stress promotes their transdifferentiation into MyoFbs [27]. Comparative analysis of RNA expression profiles between CFs and MyoFbs has identified α-SMA as a central marker and effector of transdifferentiation, serving as a hallmark of phenotypic transformation [28]. Moreover, α-SMA not only drives the transdifferentiation process but also directly mediates the upregulation of collagen synthesis (Collagen I and Collagen III) [29]. This transformation disrupts the balance of ECM metabolism, promoting the pathological synthesis and accumulation of fibrotic deposits, which ultimately contribute to the progressive deterioration of cardiac function. Hyp, an amino acid unique to animal collagen and rarely found in other proteins, serves as an indirect indicator of fibrosis severity. This study shows that after ISO induction, the Hyp content in cell culture medium, as well as the expression levels of Collagen I, Collagen III, and α-SMA proteins, were significantly elevated, confirming the successful establishment of the myocardial fibrosis model. Intervention with MG-EWE and its active components, Laurolitsine and Hecogenin, resulted in significant reductions in Hyp secretion by rat CFs, along with marked downregulation of the protein expression levels of Collagen I, Collagen III, and α-SMA, suggesting their effective inhibition of CFs transdifferentiation into MyoFbs. In conjunction with the results of the cell migration assay, MG-EWE and its active components were shown to significantly impede ISO-induced CFs migration, further demonstrating their inhibitory effects on fibroblast recruitment to the site of injury. Collectively, these results provide compelling evidence that MG-EWE and its active components possess potent anti-myocardial fibrosis capabilities.

Myocardial fibrosis is intricately linked to chronic inflammation, in which IL-6 serves as a pivotal pro-inflammatory cytokine contributing to the pathogenesis of myocardial fibrosis. Cardiac fibroblasts are key drivers and maintainers of this pathological process through the regulation of IL-6 production [30]. Studies have demonstrated that IL-6 promotes the proliferation, differentiation, and collagen deposition of myofibroblasts, with its profibrotic effects primarily mediated through the activation of the signal transducer and activator of transcription 3 (STAT3) signaling pathway in cardiac fibroblasts, thereby driving collagen synthesis [31,32]. Experimental evidence supports this mechanism: in models of left ventricular pressure overload [33,34] and diabetic cardiomyopathy [35], genetic deletion of IL-6 significantly alleviates myocardial fibrosis and associated dysfunction, whereas IL-6 overexpression exacerbates the fibrotic phenotype. The results of Real-Time Fluorescence Quantitative PCR (qPCR) and Western blot analyses in the present study reveal that MG-EWE and its active components, Laurolitsine and Hecogenin, significantly reduce both mRNA and protein expression levels of IL-6. These findings indicate that MG-EWE and its active constituents can effectively block the inflammation-fibrosis response mediated by IL-6 at its source.

The MAPK signaling pathway is intimately associated with various cellular processes, including proliferation, differentiation, migration, aging, and apoptosis [36]. It comprises three primary branches: the extracellular signal-regulated kinases 1 and 2 (ERK1/2), c-Jun N-terminal kinases 1, 2, and 3 (JNK1/2/3), and p38 MAPK signaling pathways [37]. Research has shown that abnormal activation of key branches of the MAPK pathway, particularly ERK1/2 and JNK1/2/3, is extensively involved in cardiac fibroblast proliferation and collagen maturation processes, thereby accelerating the development of myocardial fibrosis [38,39]. In this study, the biological processes identified by GO analysis were significantly enriched in the JNK cascade, and the key target JNK determined by molecular docking suggest that the MAPK/JNK signaling pathway is highly relevant to this study. The MAPK/JNK cascade operates through a conserved three-tiered kinase cascade mechanism: upstream activator proteins bind to specific receptors, leading to the sequential activation of MAP kinase kinase kinase (MAPKKK), which then phosphorylates and activates MAP kinase kinase (MAPKK, MKK4 and MKK7). This, in turn, phosphorylates and activates the JNK (including JNK1, JNK2, and JNK3 subtypes) [40,41]. Activated JNK phosphorylates members of the Jun protein family (including JunB, JunD, and c-Jun), facilitating their dimerization with Fos proteins (e.g., c-Fos, FosB, Fra-1/2) to form the transcription factor activation protein-1 (AP-1). This complex subsequently activates the transcriptional program of target genes [42,43]. Studies have indicated that the IL-6 promoter region in human fibroblasts contains AP-1 response elements [44], suggesting that the MAPK/JNK signaling pathway may drive IL-6 production in CFs via the AP-1 transcription factor. Furthermore, AP-1 has been confirmed as a necessary factor for β-adrenergic receptor-mediated IL-6 expression in cardiomyocytes. This study demonstrated that Hecogenin significantly inhibits the phosphorylation of both c-Jun, a vital member of the AP-1 transcription factor family, and its upstream kinase JNK. This inhibition ultimately results in a substantial reduction in the production of the downstream inflammatory cytokine IL-6. Under β-adrenergic receptor activation, the formation and release of IL-6 in cardiomyocytes depend on the activation of AP-1 binding sites and cAMP response elements (CRE) within its promoter, further implicating that β-adrenergic signaling may induce IL-6 production by regulating AP-1 activity [45]. Moreover, research has demonstrated that ADRB2 antagonists can inhibit cell proliferation, invasion, and metastasis by suppressing the MAPK signaling pathway and the activity of transcription factors such as NF-κB, AP-1, CREB, and STAT3 [37]. Molecular docking revealed that Laurolitsine exhibits high binding affinity to ADRB2. Consistent with this, Western blot analysis demonstrated that Laurolitsine significantly reduced the levels of p-ADRB2/ADRB2 and p-JNK/JNK, indicating inhibition of ADRB2 and JNK phosphorylation, thus suppressing the MAPK signaling pathway. These findings indicate that ADRB2 has the potential to regulate the MAPK signaling pathway, suggesting that in CFs, ADRB2 may promote AP-1 transcription factor activity by activating the JNK/c-Jun pathway, thereby regulating IL-6 production. In this study, MG-EWE, Laurolitsine, and Hecogenin, significantly reduced ISO-induced *IL-6* mRNA expression levels in rat CFs, while concurrently downregulating the phosphorylation levels of ADRB2, JNK, and c-Jun proteins, as well as IL-6 protein expression levels. These results indicate that MG-EWE and its active components, Laurolitsine and Hecogenin, can inhibit the ADRB2/JNK/c-Jun signaling pathway, thereby suppressing IL-6 transcription and protein expression in rat CFs (Figure 11). Previous research has demonstrated that low-dose ethanol can mitigate the development of myocardial fibrosis in diabetic rats by suppressing the JNK signaling pathway [46]. Additionally, knockout of the endoplasmic reticulum protein TXNDC5 has been shown to inhibit JNK pathway activation, resulting in decreased expression of fibrosis-associated proteins, reduced myocardial fibrosis, and improved cardiac functional performance [47]. Furthermore, 2-aminopyridine-4-carboxamide (2-APQC), an activator of SIRT3, significantly attenuates myocardial fibrosis by inhibiting the JNK pathway and enhancing mitochondrial function [48]. These findings collectively underscore the therapeutic potential of targeting the JNK signaling pathway for the prevention and reversal of myocardial fibrosis.

## 5. Conclusions and Future Directions

In summary, this study, based on network pharmacology and cell experiments, elucidates the mechanism by which MG-EWE and its active components, Laurolitsine and Hecogenin, inhibit CF IL-6 production through the ADRB2/JNK/c-Jun pathway. The study further confirms their ability to effectively suppress CF transdifferentiation into MyoFbs, thereby exerting an anti-cardiac fibrosis effect. Our findings demonstrate that both Laurolitsine and Hecogenin independently inhibit cardiac fibroblast transdifferentiation and IL-6 production via the ADRB2/JNK/c-Jun pathway; however, their potential synergistic effects remain to be investigated. It is essential for elucidating the integrated mechanism underlying the anti-fibrotic effects of the main active components in MG-EWE. Future research endeavors will incorporate animal models to further validate the roles of MG-EWE and its monomeric compounds, Laurolitsine and Hecogenin, in anti-myocardial fibrosis and IL-6 production inhibition. Additionally, return experiments will be conducted to clarify the role of the ADRB2/JNK/c-Jun pathway in MyoFbs phenotypic transformation. By focusing on cellular-level mechanism studies, this research identifies MG-EWE active components, Laurolitsine and Hecogenin, as promising candidate compounds for inhibiting CFs IL-6 production, demonstrating considerable potential for drug development. This study preliminarily confirms the effects of MG on inhibiting CFs transdifferentiation and IL-6 production and screens out candidate compounds for inhibiting CFs IL-6 production. This discovery not only enhances the theoretical understanding of the pharmacological effects of the traditional ethnic medicinal plant, *Cinnamomum migao* H.W. Li, in the prevention and treatment of cardiovascular diseases but may also provide a scientific foundation for developing drugs that target IL-6 production and regulate CFs transdifferentiation into MyoFbs to combat myocardial fibrosis.

## Figures and Tables

**Figure 1 cimb-47-00798-f001:**
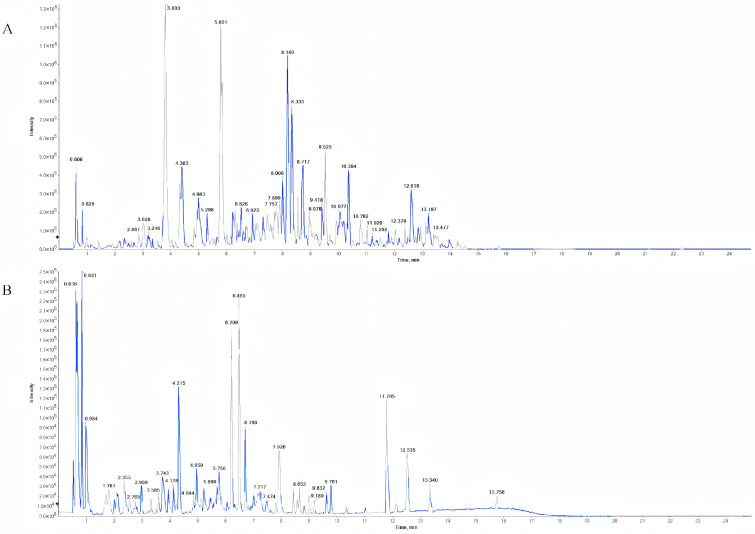
Total ion chromatograms of MG-EWE obtained by UPLC-Q-TOF-MS. (**A**) Positive ion mode total ion chromatogram; (**B**) Negative ion mode total ion chromatogram.

**Figure 2 cimb-47-00798-f002:**
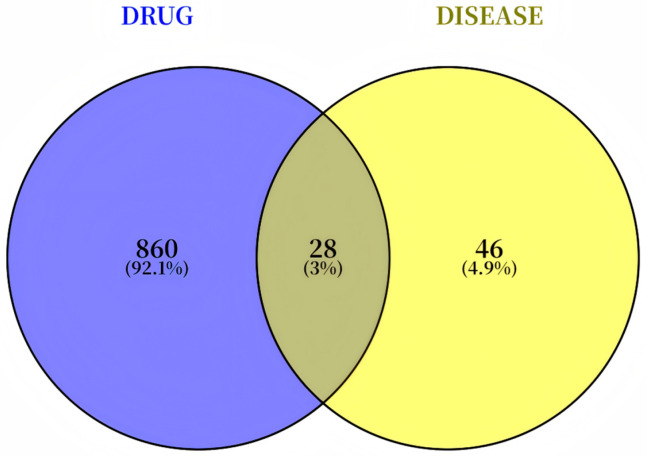
Venn diagram of MG-EWE active ingredient targets and proteins related to IL-6 production in cardiac fibroblasts.

**Figure 3 cimb-47-00798-f003:**
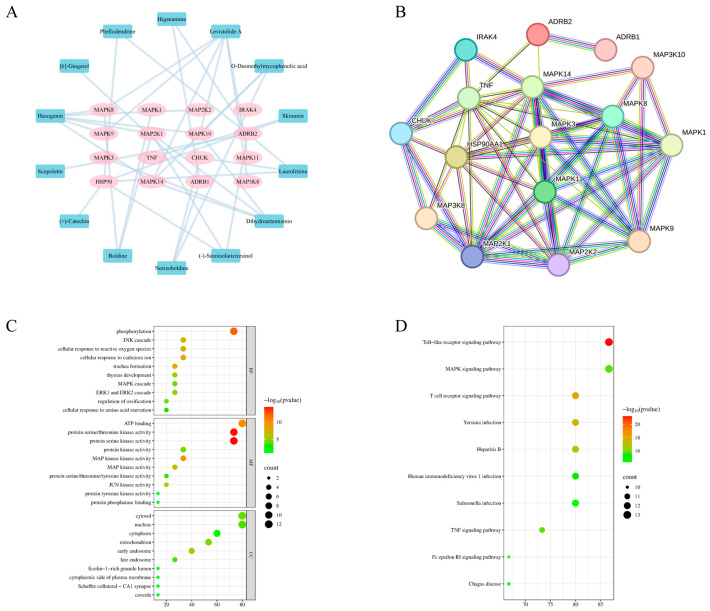
Target interaction network and gene function enrichment analysis diagram. (**A**) Compound-target interaction network diagram; (**B**) Protein–protein interaction (PPI) network diagram; (**C**) Gene Ontology (GO) enrichment analysis bubble chart; (**D**) Kyoto Encyclopedia of Genes and Genomes (KEGG) enrichment analysis bubble chart.

**Figure 4 cimb-47-00798-f004:**
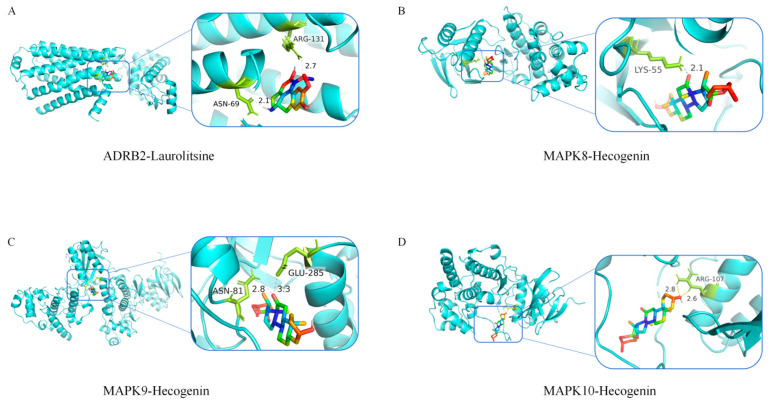
Molecular docking visualization. (**A**) Molecular docking mode diagram of Laurolitsine with ADRB2; (**B**) Molecular docking mode diagram of Hecogenin with MAPK8; (**C**) Molecular docking mode diagram of Hecogenin with MAPK9; (**D**) Molecular docking mode diagram of Hecogenin with MAPK10.

**Figure 5 cimb-47-00798-f005:**
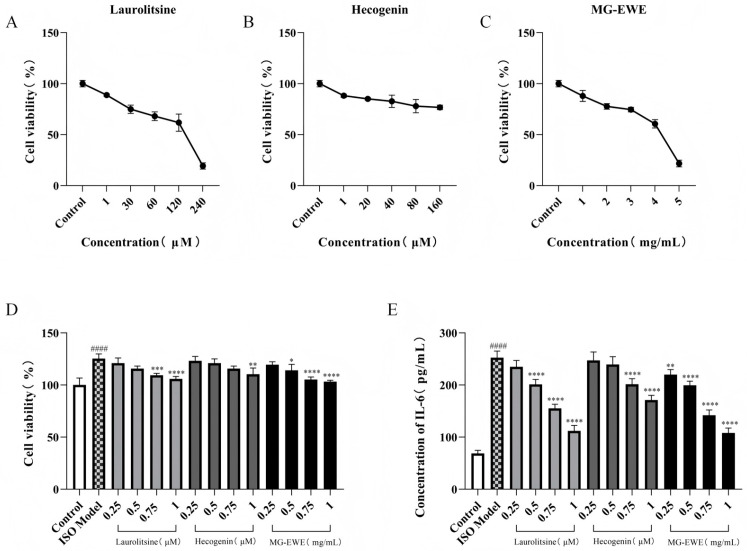
Screening of the optimal intervention concentrations of MG-EWE, Laurolitsine, and Hecogenin. (**A**) Effects of different concentrations of Laurolitsine on CFs viability; (**B**) Effect of different concentrations of Hecogenin on CFs viability; (**C**) Effect of different concentrations of MG-EWE on CFs viability; (**D**) Effects of different concentrations of MG-EWE, Laurolitsine, and Hecogenin on ISO-induced CFs viability; (**E**) IL-6 content in the cell supernatant of each group. Compared to the control group, ^####^ *p* < 0.0001. Compared to the ISO model group, * *p* < 0.05, ** *p* < 0.01, *** *p* < 0.001, **** *p* < 0.0001.

**Figure 6 cimb-47-00798-f006:**
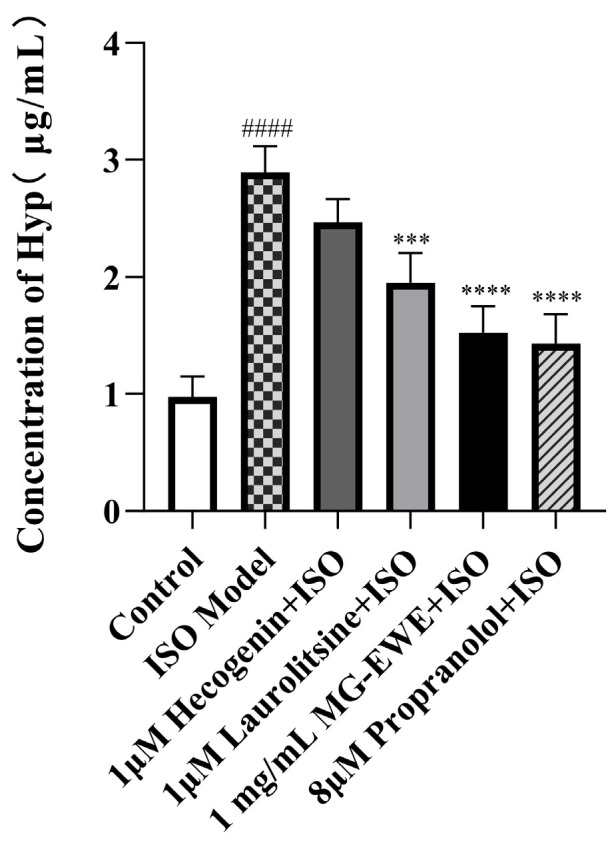
The concentrations of Hyp in the supernatant of cells in each group. Compared to the control group, ^####^ *p* < 0.0001. Compared to the ISO model group, *** *p* < 0.001, **** *p* < 0.0001.

**Figure 7 cimb-47-00798-f007:**
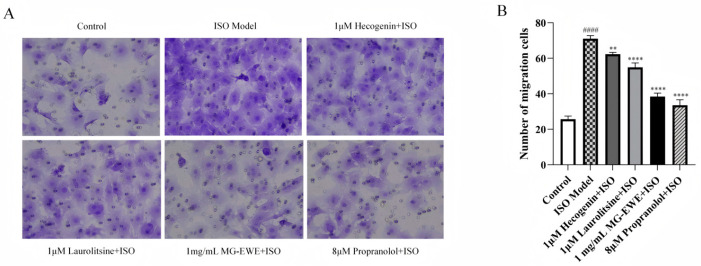
The Migration ability of CFs in each group, original magnification: 200×. (**A**) Representative images from the Transwell assay showing the migration of CFs treated with MG-EWE, Laurolitsine, and Hecogenin; (**B**) Quantitative analysis of the effects of MG-EWE, Laurolitsine, and Hecogenin on the number of ISO-induced CFs migration. Compared to the control group, ^####^ *p* < 0.0001. Compared to the ISO model group, ** *p* < 0.01, **** *p* < 0.0001.

**Figure 8 cimb-47-00798-f008:**
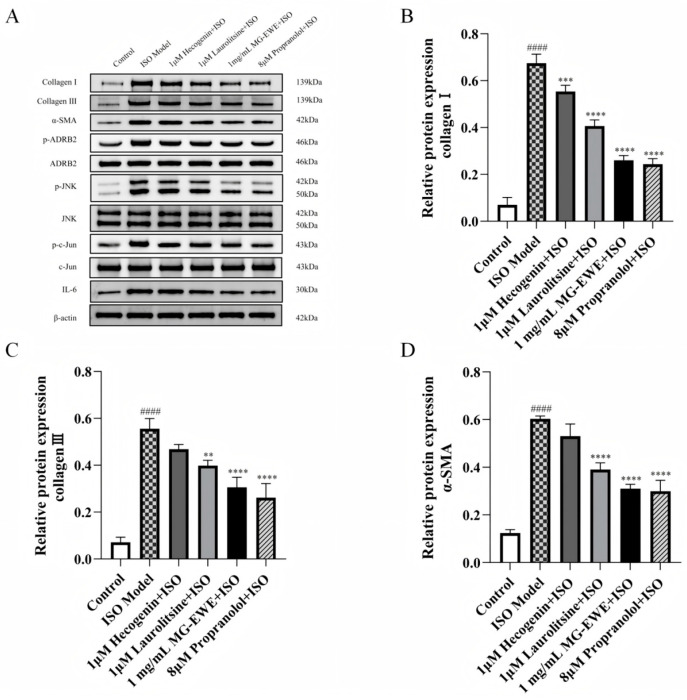
Protein expression of Collagen I, Collagen III and α-SMA in CFs of rats in each group. (**A**) Electrophoretic bands of Collagen I, Collagen III, α-SMA, p-ADRB2, ADRB2, p-JNK, JNK, p-c-Jun, c-Jun, and IL-6 protein expression; (**B**) Relative protein expression level of Collagen I; (**C**) Relative protein expression level of Collagen III; (**D**) Relative protein expression level of α-SMA. Compared to the control group, ^####^ *p* < 0.0001. Compared to the ISO model group, ** *p* < 0.01, *** *p* < 0.001, **** *p* < 0.0001.

**Figure 9 cimb-47-00798-f009:**
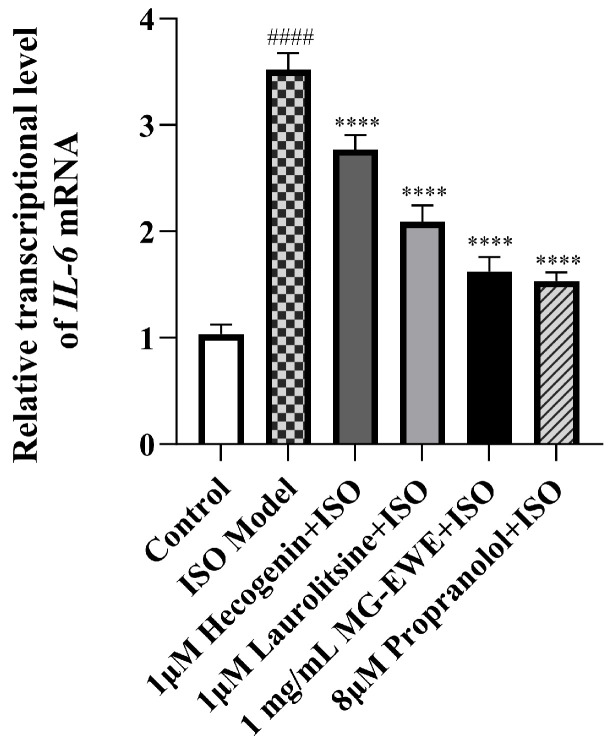
Expression level of *IL-6* mRMA in CFs of rats in each group. Compared to the control group, ^####^ *p* < 0.0001. Compared to the ISO model group, **** *p* < 0.0001.

**Figure 10 cimb-47-00798-f010:**
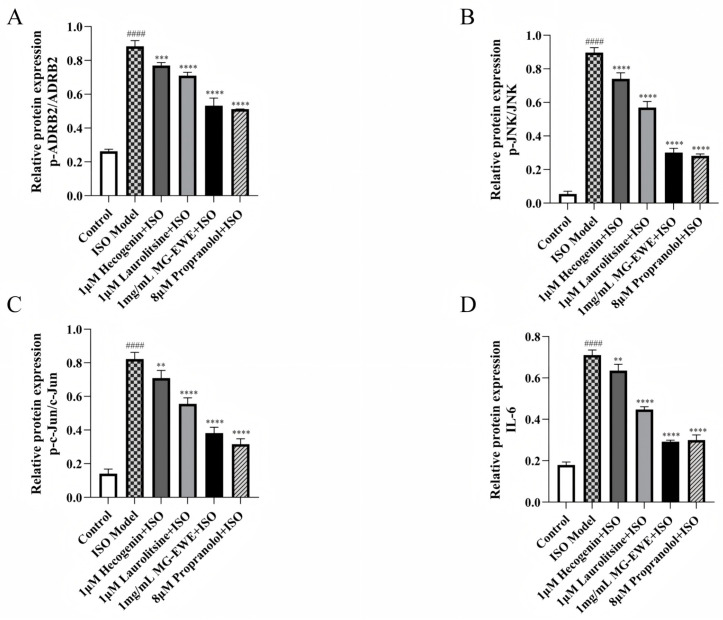
Protein expression levels of ADRB2/JNK/c-Jun signaling pathway-related proteins and IL-6 in CFs of each group. (**A**) Relative expression levels of p-ADRB2; (**B**) Relative expression levels of p-JNK; (**C**) Relative expression levels of p-c-Jun; (**D**) Relative expression levels of IL-6. Compared with the Control group, ^####^ *p* < 0.001. Compared with the ISO Model group, ** *p* < 0.01, *** *p* < 0.001, **** *p* < 0.0001.

**Figure 11 cimb-47-00798-f011:**
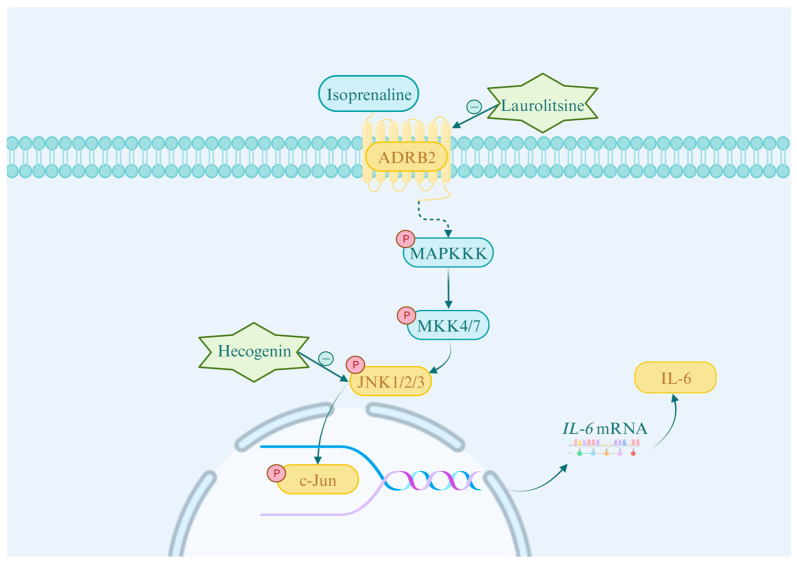
The schematic diagram illustrates the molecular mechanism through which MG suppresses IL-6 production in cardiac fibroblasts (CFs): in the isoproterenol (ISO)-induced CFs transdifferentiation model, MG-EWE and its active components Laurolitsine and Hecogenin can inhibit the abnormal activation of the ADRB2/JNK/c-Jun signaling pathway. When ISO binds to the ADRB2 receptor on the cell membrane surface of CFs, it initiates a downstream signaling cascade. Activation of this receptor leads to the phosphorylation of key signaling molecules, including JNK1/2/3 and c-Jun, thereby activating the JNK/c-Jun signaling pathway. The activated JNK1/2/3 translocates from the cytoplasm to the nucleus and further phosphorylates the transcription factor c-Jun within the nucleus, enhancing its transcriptional activity. This ultimately promotes the transcription of the *IL-6* gene and increases its protein expression. Laurolitsine functions by inhibiting the activation of ADRB2 on the CFs cell membrane surface, while Hecogenin inhibits the phosphorylation of JNK, thereby blocking the subsequent phosphorylation of c-Jun. Together, these actions result in the joint inhibition of IL-6 production.

**Table 1 cimb-47-00798-t001:** List of primer designs.

Gene	Forward	Reverse
*IL-6*	CGATGATGCACTGTCAGAAAAC	ACTCCAGGTAGAAACGGAACTC
β-actin	TGACGTTGACATCCGTAAAGACC	GTGCTAGGAGCCAGGGCAGTAA

**Table 2 cimb-47-00798-t002:** Molecular docking binding energy of MG-EWE and core targets.

Ingredients	CAS Number	Targets	Score (kcal/mol)
Laurolitsine	5890-18-6	ADRB2	−10.2
Hecogenin	467-55-0	MAPK9	−10.1
Skimmin	93-39-0	HSP90	−9.9
Hecogenin	467-55-0	MAPK14	−9.5
Levistolide A	88182-33-6	MAPK11	−9.5
Higenamine	5843-65-2	ADRB2	−9.4
Hecogenin	467-55-0	MAPK8	−9.4
(+)-Catechin	154-23-4	HSP90	−9.4
Norisoboldine	23599-69-1	IRAK4	−9.3
Dihydroartemisinin	71939-50-9	MAPK14	−9.1
Phellodendrine	104112-82-5	HSP90	−9.1
Norisoboldine	23599-69-1	CHUK	−9.0
Boldine	476-70-0	MAP2K1	−9.0
Levistolide A	88182-33-6	MAPK14	−8.9
Hecogenin	467-55-0	MAP2K2	−8.9
(−)-Secoisolariciresinol	29388-59-8	HSP90	−8.8
Higenamine	5843-65-2	ADRB1	−8.7
Hecogenin	467-55-0	MAP2K1	−8.7
Hecogenin	467-55-0	MAPK10	−8.7
O-Desmethylmycophenolic acid	31858-65-8	MAPK10	−8.7
Levistolide A	88182-33-6	MAP3K8	−8.7
Dihydroartemisinin	71939-50-9	MAPK3	−8.7
Dihydroartemisinin	71939-50-9	MAPK10	−8.6
Levistolide A	88182-33-6	MAPK8	−8.5
[6]-Gingerol	23513-14-6	ADRB1	−8.4
Laurolitsine	5890-18-6	CHUK	−8.4
(−)-Secoisolariciresinol	29388-59-8	MAP2K1	−8.4
Dihydroartemisinin	71939-50-9	MAP2K1	−8.4
Levistolide A	88182-33-6	MAPK9	−8.4
Boldine	476-70-0	MAPK1	−8.3
Boldine	476-70-0	MAPK8	−8.3
O-Desmethylmycophenolic acid	31858-65-8	IRAK4	−8.3
Laurolitsine	5890-18-6	HSP90	−8.3
(−)-Secoisolariciresinol	29388-59-8	ADRB2	−8.2
Phellodendrine	104112-82-5	ADRB2	−8.2
Scopoletin	92-61-5	ADRB2	−8.2
Skimmin	93-39-0	TNF	−8.2
O-Desmethylmycophenolic acid	31858-65-8	MAPK14	−8.2
Levistolide A	88182-33-6	MAPK10	−8.2
Norisoboldine	23599-69-1	ADRB1	−8.0

## Data Availability

The original contributions presented in this study are included in the article. Further inquiries can be directed to the corresponding author.

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
