# Peer review of "Cinnamomum migao H.W. Li Ethanol-Water Extract Suppresses IL-6 Production in Cardiac Fibroblasts: Mechanisms Elucidated via UPLC-Q-TOF-MS, Network Pharmacology, and Experimental Assays"

_cimb, 2025, doi:10.3390/cimb47100798_

Round 1
Reviewer 1 Report
Comments and Suggestions for Authors
I am very honored to be invited to review the manuscript entitled “Cinnamomum migao H.W. Li Ethanol-water Extract Suppresses IL-6 Production in Cardiac Fibroblasts: Mechanisms Elucidated via UPLC-Q-TOF-MS, Network Pharmacology, and Experi-mental Assays”. This study provides a comprehensive and systematic investigation into the mechanisms by which the ethanol-water extract of Cinnamomum migao H.W. Li (MG), specifically referred to as MG-EWE, suppresses the transdifferentiation and interleukin-6 (IL-6) production in cardiac fibroblasts through multiple biological pathways. The research team innovatively combined network pharmacology prediction with experimental validation, successfully identifying two active components, Laurolitsine and Hecogenin, in MG-EWE, and further confirmed that MG-EWE may intervene in the process of ISO-induced IL-6 production in cardiac fibroblasts through the ADRB2/JNK/c-Jun signaling pathway.
From a pathophysiological perspective, when the heart is subjected to adverse stimuli such as myocardial infarction, cardiac fibroblasts are activated, accompanied by a significant increase in IL-6 expression levels. This pro-inflammatory cytokine not only directly promotes the proliferation, activation, and collagen synthesis of cardiac fibroblasts, thereby exacerbating the degree of fibrosis in the infarcted area; but also, IL-6 released by fibroblasts further intensifies the local inflammatory response and damages cardiomyocytes, forming a vicious cycle of "inflammation-fibrosis-cardiomyocyte injury." Notably, although most current studies focus on the transdifferentiation process of cardiac fibroblasts, abnormal deposition of extracellular matrix, and cardiomyocyte injury, the critical role of IL-6 generated by cardiac fibroblasts in the process of myocardial fibrosis has not received sufficient attention. Cinnamomum migao H.W. Li (MG) is a traditional herb widely used in the Miao ethnic regions of China, primarily for treating cardiovascular and cerebrovascular diseases. This study takes a different approach, focusing on the generation mechanism of IL-6 in cardiac fibroblasts, and systematically investigates the pharmacological effects of MG for the first time. Through in-depth research, active components capable of effectively inhibiting IL-6 generation were successfully screened. This important discovery not only provides new insights for the development of therapeutic drugs for myocardial fibrosis but also lays the foundation for the discovery of lead compounds, demonstrating considerable innovative value in the related field. The research topic aligns with the scope of this journal, the full text presents a rigorous logical structure, the experimental design is reasonable, the data arguments are sufficient, and the language expression is accurate and fluent, showing high application prospects.
However, there are still some deficiencies, as follows:
- The introduction mentions “Modern pharmacological research indicates that MG can attenuate isoproterenol (ISO)-induced myocardial injury and alleviate symptoms of heart failure in rat models”. This statement lacks corresponding reference support. It is recommended to supplement the source of this citation to enhance the credibility and academic rigor of the statement.
- The introduction provides a relatively brief overview of the pharmacological research progress of MG and its application in clinical drugs. If relevant studies have a certain foundation, it is recommended to further supplement the related content to enhance the academic background support of the introduction.
- The article states that the mechanism of action of MG-EWE is achieved through the “synergistic regulation” of two monomers (Laurolitsine and Hecogenin). However, the current results section only presents experimental data under separate treatment conditions of MG-EWE, Laurolitsine, and Hecogenin, and does not provide an experimental design or data proving the synergistic effect between these two monomers. Therefore, it is recommended to appropriately revise the expression of “synergistic regulation” to align with the actual research content.
- Some professional terms or specific words in the article are not italicized as per the standard, such as botanical names or gene names. In addition, the font format in the caption of Figure 8 is inconsistent. It is recommended to carefully check the entire article to ensure that the relevant formats comply with academic writing standards.
Reviewer 2 Report
Comments and Suggestions for Authors
This article covers important role of Cinnamomun migao H.W. Li (MG-EWE) ethanol-water extract inhibiting cardiac fibroblast (CF) and IL-6 biosynthesis and providing insights into its anti-myocardial fibrosis effects.
This specific strategy presented in this article is designed to underlying molecular mechanism by analyzing a total of 173 compounds identified in MG-EWE. Network pharmacology analysis was used to identify active constituents and their mechanism in inhibiting IL-6 synthesis in CFs.
The compiled data are supported with 11 important figures and 2 tables. The article concludes with 41 very recent literature references and supplementary materials. This consolidated study constitutes crucially important developments, which were never ever reported and in such systematic and specific order and sequences.
The following suggested changes and recommendations should be introduced before the publication of the manuscript:
- Page 2, Multiple marks [Error!] Must be fixed. No literature references are included.
- Page 19, Figure 11. Should be inserted earlier in the Discussion section, preferably in line 510, discussing MAPK signaling pathway. This section should be expanded to address other important interactions including specific Laurolitsine functions.
- Page 19, Line 634 This paragraph should constitute missing section of Conclusions. This section must be split in two sections such as Perspective and Conclusions and highlight the specific experiments to verify the outcome of this work. In its present format, the authors do not fully describe the desired/anticipated effects without citing important literature references. Authors should also include comparative data available in the literature regarding previous information on the role of other molecules inhibiting Il-6 production through synergistic regulation of the ADRB2/JNK/c-Jun signaling and their potential role for drug development targeting myocardial fibrosis.
The manuscript is of good quality, and written to meets the standard for articles published in Current Issues in Molecular Biology. I recommend it for publication after the correction of these important and suggested changes.
Comments on the Quality of English LanguageThis article covers important role of Cinnamomun migao H.W. Li (MG-EWE) ethanol-water extract inhibiting cardiac fibroblast (CF) and IL-6 biosynthesis and providing insights into its anti-myocardial fibrosis effects.
This specific strategy presented in this article is designed to underlying molecular mechanism by analyzing a total of 173 compounds identified in MG-EWE. Network pharmacology analysis was used to identify active constituents and their mechanism in inhibiting IL-6 synthesis in CFs.
The compiled data are supported with 11 important figures and 2 tables. The article concludes with 41 very recent literature references and supplementary materials. This consolidated study constitutes crucially important developments, which were never ever reported and in such systematic and specific order and sequences.
The following suggested changes and recommendations should be introduced before the publication of the manuscript:
- Page 2, Multiple marks [Error!] Must be fixed. No literature references are included.
- Page 19, Figure 11. Should be inserted earlier in the Discussion section, preferably in line 510, discussing MAPK signaling pathway. This section should be expanded to address other important interactions including specific Laurolitsine functions.
- Page 19, Line 634 This paragraph should constitute missing section of Conclusions. This section must be split in two sections such as Perspective and Conclusions and highlight the specific experiments to verify the outcome of this work. In its present format, the authors do not fully describe the desired/anticipated effects without citing important literature references. Authors should also include comparative data available in the literature regarding previous information on the role of other molecules inhibiting Il-6 production through synergistic regulation of the ADRB2/JNK/c-Jun signaling and their potential role for drug development targeting myocardial fibrosis.
The manuscript is of good quality, and written to meets the standard for articles published in Current Issues in Molecular Biology. I recommend it for publication after the correction of these important and suggested changes.
